# The Impact of COVID-19 on Highway Traffic and Management: The Case Study of an Operator Perspective

**Carlos Oliveira Cruz** [1,*] and **Joaquim Miranda Sarmento** [2]

1   CERIS, Instituto Superior Técnico, Universidade de Lisboa, 1649-004 Lisbon, Portugal
2   ADVANCE, Instituto Superior de Economia e Gestão, 1200-781 Lisbon, Portugal; jsarmento@iseg.ulisboa.pt
*   Correspondence: oliveira.cruz@tecnico.ulisboa.pt

**Abstract:** The COVID-19 pandemic created an enormous disruption to the everyday life of the modern society. Among the various urban systems, transportation services were among those that suffered the most significant impacts, particularly severe in the case of highways. This paper addresses the challenges and responses to the pandemic from a private highway operator's perspective and from a multidisciplinary perspective. Highway operators faced two main challenges: on one hand, the need to cope with the potential disruption caused by the pandemic and a national lockdown for almost three months, the provision of road services, and the requirement to ensure the proper operation and maintenance, and on the other hand, the strong negative impact of the pandemic on levels of traffic. Our case study shows that the operator's management response in question is essentially characterised by being a first response to short term impacts while balancing for workers health and safety, engineering and management, internal business management, and overall economic impact. Highway operators were hardly prepared for such an event and became more focused on prioritising their employees and clients' safety to avoid service disruption. Regarding levels of traffic, the pandemic has had severe effects, although to a varying degree, depending on the different types of vehicles (heavy, light, passenger, freight, among other types of vehicles) and the location of highways (coastal vs. interior). The lessons learnt can be valuable in future disruptive events and for other highway concession operators.

**Keywords:** COVID-19; highway management; variation in levels of traffic

## 1. Introduction

In 2020, the world was surprised by the first global pandemic of the XXI century, the new Coronavirus caused, the origin of a disease called COVID-19, or SARS-Cov 2 [1]. The virus appeared in the Chinese city of Wuhan at the end of 2019 and quickly spread over the world in a few months. By March 2020, COVID-19 had hit most countries globally [2]. To control this pandemic, most countries imposed what the IMF called the "great lockdown" [3] from March to June of the same year. This meant closing most economic activities and non-essential business (such as hotels, restaurants, and the majority of shops). Only enterprises such as supermarkets or pharmacies and other health services were allowed to remain open, as they were considered essential services, albeit with several restrictions and safety measures. Schools were also closed, and public gatherings (both at events and in parks) were banned. Naturally, these measures also included a substantial restriction in travelling, especially by air, rail and public transportation, given the high concentration of people and the increase in probability of spreading the virus. These quarantine measures (together with self-quarantine for the majority of people and firms and social distancing) resulted in most employees working from home, increasing the percentage of teleworking to previously unknown levels [4].

This pandemic produced an unprecedented economic crisis, with most countries experiencing a two-digit drop in GDP during the 2nd Quarter and the overall year [5]

Furthermore, most of the economic institutions forecast a slow recovery over the next few years [6], although much uncertainty is still present, as there is a lack of prediction regarding the evolution of the diseases and the discovery of a vaccine [7].

All of these safety measures led to a substantial decline in transportation during these months ([8]. In particular, the level of road traffic has dropped substantially, partly because people had to work from home virtually and because leisure and travelling were prohibited during those months. However, unlike air traffic, which fell by more than 90%, the decline in road traffic was lower [9]. This was due to three main reasons. The first was because certain workers (such as health workers or those employed by the providers of essential products or services) continued to go to work and continued to drive on the roads. A second reason was logistics and distribution, which required an increase in the door-to-door distribution to tackle online shopping's exponential growth [10,11]. The third reason was that there was a move away from public transport to the use of private cars in the majority of cases. Despite the recent increase in the use of public transport in the previous years (see, for instance, [12]), the pandemic has led to a substantial decline in such transportation modes. This was not just for safety reasons but was also a consequence of the lack of public transport due to the pandemic restrictions [13]. On the contrary, a proportion of freight traffic continued to circulate, as the essential businesses had to be supplied with goods.

Nevertheless, the pandemic and the substantial drop in road traffic levels, particularly highway traffic, raised new challenges for the management and concession of those highways [14]. The complexity and interdisciplinary nature of these challenges have no parallel for road concessionaires, which, on its own, raises several new topics for research. For instance, the difficulty in managing the trade-off of ensuring that the operation remains unrestricted, considering the structural importance of the road system, but, at the same time, ensuring the safety and well-being of works in a sector where face-to-face interaction might be inevitable, for example, in road assistance activities.

We followed research based on a case study methodology. As [15] referred to, case studies are helpful to introduce new theory based on empirical data. The case study method is used in many scientific works concerning the transport sector, for example [11–13,16,17].

This is what is provided in this paper. Facing a new and unexpected global event, the Coronavirus pandemic, we used the Portuguese highway operator to assess the following research questions:

(1) How did the highway operator respond to the pandemic crisis and to the need to keep operation activities and health safety measures?
(2) What was the impact on traffic levels (by type of sector, vehicles, and location of motorways—rural vs. urban, interior vs. coastal)?

Using this case study, we can confront how a large highway operator could face an emergency not foreseen in contingency plans. We also analyse how traffic evolved after the government introduced a strong and broader "lockdown", imposing extensive restrictions on people's mobility. Following [18], we used this case study to provide exploratory work and explanatory variables. Using data on highway traffic, it is possible to assess not only the global impact of a "lockdown", but also the impacts at sector level, as trucks and freights in some economic activities were allowed to continue to operate, vs. the commercial vehicles of the individual, mainly restricted by confinement. The experience of the Portuguese operator allows for an almost "laboratory experience", a "natural experiment", in the words of [19], that can help to extract future recommendations and lessons for other road concessionaires.

Our research shows that the pandemic led to a focus on providing a short-term response to ensure the continuity of the provision of services. However, in terms of traffic, the evidence pointed to a significant decrease in traffic levels (40% in coastal highways and 38% in interior highways), particularly for light passenger vehicles, but to a less extent for trucks and commercial vehicles.

Understanding the impact of COVID-19 on road concessions management and operation requires an interdisciplinary approach based on the following areas: health and safety, engineering, business and management, and economic impact.

This paper is organised as follows: Section 2 presents the case study research and methodology. The response of the highway operator to the pandemics is described and analysed in Section 3. The impact on traffic and revenues as well as on accidents is detailed in Section 4. Section 5 presents the discussion section from an interdisciplinary perspective. Finally, Section 6 concludes.

## 2. Case Study Research and Methodology

Portugal registered the first case of COVID-19 on 2 March 2020, and the first death on March 16. The government imposed a lockdown, similar to those decreed in most European countries, between mid-March and the end of May. The lockdown in Portugal was a one common-level, meaning that rules and restrictions were equal across the whole country. This lockdown was nation based, covering all of the country with the same type of measures and with the same degree of restrictions. This led to a significant economic downturn, with recent statistics showing a year-on-year drop of 3.8% in GDP in the 1st Quarter and 16.6% in the 2nd Quarter. The highway traffic level dropped by 50% in the 1st Semester, with a drop of 80% in April.

In this paper, we use the case of BRISA, the largest highway operator in Portugal, which manages 1600 km and 16 highways concessions [20,21]; (see Figure 1 for a map of these concessions). We had access to first-hand information on how the firm responded to this crisis and data regarding the pandemic's impact during March, April, and May, in terms of traffic, revenues, and accidents.

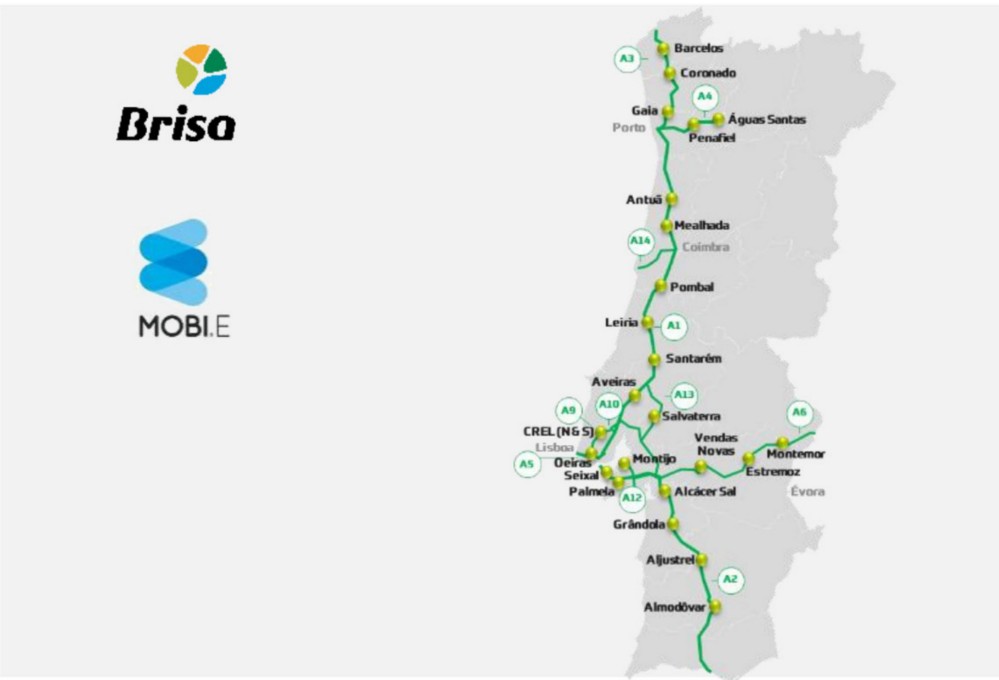

**Figure 1.** Brisa highways concession map. Source: www.brisa.pt (accessed on 2 March 2021).

We used a case study methodology to address fundamental questions in times of pandemics. Our first research question is "how did the highway operator respond to the pandemic crisis and to the need to keep operation activities and health safety measures?". This is a novel question, as no highway operators' contingence plans had precedented such an event [22,23]. Therefore, highway operators faced a new, unexpected, and dire situation with very little time to respond effectively. Between the first case of the new Coronavirus

and the lockdown, there were only ten days. We used this case study to address a novel question as referred by [15,16].

Following this, we also aim to analyse a second research question: what was the impact of the quarantine and lockdown measures on traffic levels? By collecting data on the highway traffic before, during, and after the lockdown, we can assess an explanatory construction of the different impacts of this event. These different impacts were according to economic activities and commercial vehicles of individuals, more forced to confinement.

Therefore, we need to build theories from case study research [24]. This has to be done with testable constructs that can be readily measured and the hypothesis that can be tested in order for the resultant theory to be empirically valid. We need to employ different levels of analysis and data, both qualitative and quantitative. Considering the various uncertainties and unknowns about this virus and its impacts, it is critical to understand opinions and suggestions from experts within the transport sector and related planning fields [23].

This paper used a qualitative approach for our first research question to assess how BRISA faced a new, unexpected, serious situation requiring a fast and robust response. The description of BRISA management's actions and decisions allows us to understand how the firm could keep operations under severe health measures and with most of the labour force working from home. As this is not possible for several operating activities, such as manual collecting of tolls in each highway exit (with hundreds of exits along with more than 1000 km) or road assistance, the firm was placed under enormous difficulties to balance the need for staff to work in person and the urge to protect the health of staff and users.

We then turn to a quantitative approach to answer our second research question, "what was the impact on traffic levels by type of sector and vehicles". The restrictive measures taken in Portugal, but also any other country, were asymmetric. Several sectors were completely shut down, while others continued to operate [25]. We collected data on highway traffic before, during, and after the lockdown. As we were able to collect data on 12 different highways, we can assess several impacts. First, the impact on different types of vehicles, ranging from commercial vehicles used by individuals, to large trucks and freight used by firms. Second, the impact on highways in coastal (urban) areas vs. in interior (rural) areas. Third, how fast the drop in traffic was, compared with the recovering period after the lockdown. Fourth, if the drop and recovery were different between urban and rural areas.

Table 1 provides a brief description of each highway. Twelve highways were included in the study, as even though BRISA operates a total of 16 highways, 4 of those highways are BCR (Brisa Concessão Rodoviária). We classified each highway as either located in Portugal's coastal region or the interior [26]. Portugal's coastal part is mainly urban and is heavily populated (more than 80% of Portugal's population is concentrated there), more developed, and is where almost all the country's industry is concentrated. In contrast, the interior is mainly rural, with less population and a low concentration of industry. Our study considers that seven highways are located in the coastal region and five in the interior region. Accordingly, our study aimed to both analyse the variation in levels of traffic during the period of March to June for each highway and compare the results for the highways in the coastal region with those located in the interior region. We expect that there would indeed be differences between these two regional groups.

This essentially measures the constructs and verifies the relationships between them. Case study hypotheses emerge from data analysis, rather than being made a priori, and emerge only after a careful comparison of data and constructs [24]. Based on this, we built four hypotheses for our data and analysis:

**Hypothesis 1 (H1).** *The reduction in traffic in individual vehicles is much higher than in firms' commercial vehicles, particularly in heavier vehicles.*

**Hypothesis 2 (H2).** *The reduction in traffic is much more substantial in urban areas, where more people are working in services, and are therefore able to work remotely from home, than in rural areas, where people work in agriculture and light industry, where remote work is usually not possible.*
**Hypothesis 3 (H3).** *The reduction in traffic was swift at the beginning of the lockdown, but recovery is substantially slow.*
**Hypothesis 4 (H4).** *The reduction in traffic is faster in urban areas, and the recovery is slower in such areas.*

**Table 1.** Description of highways.

| Highway | Length (km) | I/C |
|---------|-------------|-----|
| A1 | 297 | C |
| A2 | 236 | C |
| A3 | 113 | C |
| A4 | 51 | I |
| A5 | 25 | C |
| A6 | 156 | I |
| A9 | 35 | C |
| A10 | 40 | I |
| A12 | 29 | C |
| A13 | 79 | I |
| A14 | 40 | C |
| BCR | 167 | I |

Source: Firm reports. Note: I represent highways mainly located in the interior region, and C is for those mainly located in the coastal region.

## 3. Highway Management Response to the Pandemic

With regards to health issues solely, the firm assumed that it was facing two main risks: first, the transmission of the virus from one person to another in its facilities; and second, the possibility of several (or even all) of its staff being unable to work due either to illness or other non-health reasons, such as the lack of public transport, the closure of schools, and the need for staff to stay at home with their children, among other motives. Both risks were critical for ensuring the continuity of operations. Highways are a critical element, if not the main critical element, in regional mobility, ensuring the access, among others, to health service, which would be critical in the pandemic context.

However, in a highway operation, everyone cannot work from home. Back-office functions (such as accounting, payroll, marketing or human resources) can be carried out remotely, but other functions (operation and maintenance related activities) must be performed in person. Manual collection of tolls still has to be done—as not all drivers have electronic tolling—and accident assistance is an essential service for users which has to be maintained. Despite the extensive digitalisation and the high degree of automatic payments that BRISA benefits from [27], at each highway exit, there is still a need to maintain at least one manual collection of tolls which has to be in function 24/7. Additionally, the highway security staff still need to be present, for although levels of traffic have reduced substantially during the pandemic, there is still the need for automobile assistance to be available on a 24/7 basis. Finally, highway maintenance is a constant necessity, as is the monitoring of critical infrastructures, such as bridges, tunnels, and banks, even if fewer cars and trucks are using the highway network.

The firm followed the legal and medical recommendations of the National Health Authorities concerning the health and 1safety of its staff, clients, and facilities. Furthermore, it decreed the obligation for any staff member who feels that they are experiencing the symptoms of the virus to immediately report the case to both the firm and the authorities. A plan for permanent disinfection and cleaning was adopted in each facility.

Furthermore, an isolation room was also set up in each facility, where any employee not working from home should immediately move to should they feel that they are experi-

encing symptoms during working hours. Every employee who felt the symptoms outside the firm's facilities was instructed not to report for work.

As mentioned above, some staff members were unable to work from home, namely those working in functions of security, manual toll collection, and highway maintenance. The firm decided to reallocate these employees and all of the firm's facilities, whilst ensuring that all employees subject to such conditions were placed in a sufficiently large facility to accommodate all of the safety and health measures, especially those related to social distancing and the isolation area.

In parallel, all procedures were reviewed, and several measures were taken to reduce the probability of close contact between employees and clients, for example, the manual toll collection staff. To diminish contagion risk, the firm recommended that clients avoid manual payment and instead pay with electronic cards as opposed to money. Also, other safety measures were implemented, such as the wearing of gloves for receiving manual payment in coins and notes, and the automobile assistance staff were also obliged to wear personal protective equipment and to follow safety measures, mainly when providing road-side assistance, especially the maintaining of social distancing from the driver (who should remain in the car all time, if possible).

The facilities for providing front-office administrative support to clients (late payments, complaints, purchases of equipment, among others) were all closed, and this service was moved exclusively to online or telephone services.

Adopting all of these precautionary measures and actions enabled the firm to continue operating and providing all the core services without any interruption or incidents, whilst maintaining the same level of quality and service as was in place before the pandemic situation.

## 4. Impacts on Traffic and Revenues

We collected data on the levels of traffic for the highways under concession to BRISA for the first six months of 2020. On average, traffic levels were reduced (vis-à-vis with 2019) by 46% from March to June of 2020. This was due to COVID-19, as January and February had registered increases in traffic levels of 3.4% and 3.2%, respectively. Indeed, the EU and IMF pre-COVID-19 forecasts for employment, GDP, and tourism all predicted a moderate growth for the First Semester of 2020, which would have helped sustain growth in traffic levels. Figure 2 presents the year-to-year variation per month. A drop of 40% in the traffic level can be seen for March, as the "lockdown" started in the middle of the month; however, further analysis shows that during the last 15 days of the month, the level of traffic was reduced by more than 80%. The reduction in the level of traffic in April (compared with April of 2019) was around 70%, and there was a slight recuperation of traffic in May and June, which registered a reduction of 48% and 28%, respectively.

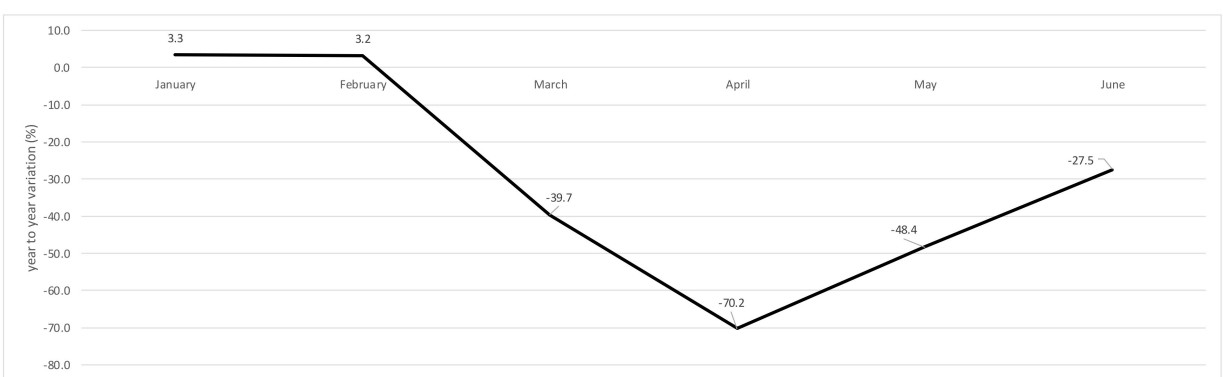

**Figure 2.** Monthly variation of traffic (year-on-year variation). Source: Authors, based on data from Brisa.

Table 2 presents the variation in traffic levels by type of vehicle from March to June 2020. We tested our first hypothesis that the reduction in traffic in individual vehicles is much higher than in firms' commercial vehicles, particularly in more heavy vehicles. As mentioned above, there was an average reduction in the levels of traffic of approximately 46% between March and June. However, this reduction was substantially different when we analyse the results by the different type of vehicles. Class 1 (which includes light passenger vehicles) registered an average reduction of almost 50%, mainly owing to the fact that, as mentioned above, most workers worked from home and the majority of leisure activities were banned. At the peak of the crisis during April, the reduction in levels of traffic of Class 1 was around 75%, which shows that the only people travelling were mainly employees of essential businesses and services, or those travelling to access basic services (pharmacies, supermarkets, among other sectors). The commercial traffic level showed less reduction, with Class 2 and Class 3 vehicles (light commercial vehicles) registering a drop in traffic levels of 35% and 22%, respectively. During the peak, in April, these types of vehicles registered a reduction of 52% for Class 2 and 33% for Class 3. When it comes to Class 4 (heavy trucks), the level of this reduction was lower, at only 13%—whereas, at the peak of the crisis in April, the equivalent reduction was approximately 24%—and there was no reduction in traffic at all for this class in March, with almost complete recovery occurring in June.

**Table 2.** Monthly variation of traffic by type of vehicle (year-to-year variation).

|  | January | February | March | April | May | June | Average Variation (March to June) |
|---|---|---|---|---|---|---|---|
| Class 1 | 3.4 | 3.5 | −43.1 | −74.5 | −50.6 | −29.1 | −49.0 |
| Class 2 | 6.0 | 3.8 | −24.9 | −52.0 | −39.5 | −22.2 | −34.7 |
| Class 3 | 4.9 | 1.5 | −7.9 | −33.3 | −31.4 | −14.4 | −22.1 |
| Class 4 | −2.1 | −4.2 | −0.1 | −23.8 | −23.6 | −2.7 | −12.8 |
| Class 5 | 7.9 | 9.5 | −48.8 | −79.4 | −51.8 | −29.6 | −49.8 |
| Average Variation | 3.4 | 3.2 | −39.8 | −70.5 | −48.5 | −27.6 | −46.3 |

Note: classes of vehicles: Class 1—Vehicles with two axles with a height measured vertically from the first axle of less than 1.10 m, with or without a trailer; Class 2—Vehicles with two axles with a height measured vertically from the first axle of equal to or greater than 1.10 m; Class 3—Vehicles with three axles with a height measured vertically from the first axle of equal to or greater than 1.10 m; Class 4—Vehicles with three or more axles with a height measured vertically from the first axle of equal to or greater than 1.10 m; Class 5—Motorcycles.

We confirmed our hypothesis that light passengers' vehicle traffic was much more reduced than other vehicles, particularly those involving transport and freight of essential goods. These values also present evidence that the supply of goods was maintained, particularly to supermarkets and pharmacies, and that growth was seen in distributing online shopping goods. Furthermore, evidence also suggests that other firms, even if closed, adopted a policy of increasing their inventories (to guarantee that materials and products would be available once the lockdown was over and production was able to start again. Data from the Portuguese Statistical Office (file:///C:/Users/JSarmento/Downloads/31CNT2T2020_B2016.pdf (accessed on 2 March 2021).) for the GDP of the 2nd Quarter of 2020 shows a substantial drop in GDP (a year-on-year reduction of 16.3%) and also a reduction in investment of 11%. Even so, there was a year-on-year increase in inventories. Of interest is the proof that a substantial increase in e-commerce and home delivery contributed significantly to these results.

The variation of levels of traffic in March of 2020 by highway classification are presented in Figure 3. We confirmed our second hypothesis, that the reduction in traffic is much more substantial in urban areas, where more people are working in services, and therefore able to work remotely from home, than in rural areas, where people work in agriculture and the light industry, where remote work is usually not possible. Even though, as mentioned by [28], there is a trend of a large modal shift from public transport to other modes, our results show that whereas coastal highways had an average reduction of 40% (which was the same level as the total reduction of our sample), the highways located in the

interior region demonstrated a slightly lower reduction of 38%. It is essential to mention that when analysing these results, one must take into consideration the effect of the A6 highway, which is an interior highway that is mainly used for transporting goods between Portugal and Spain and was thus negatively affected when the borders between the two countries (Spain is the only country with which Portugal has an inland border) were closed between mid-March and the end of June 2020. As shown in Figure 3, the other highways in the interior registered traffic levels are well below the total average. If we exclude the A6 highway, the reduction in traffic level in the interior was 36%.

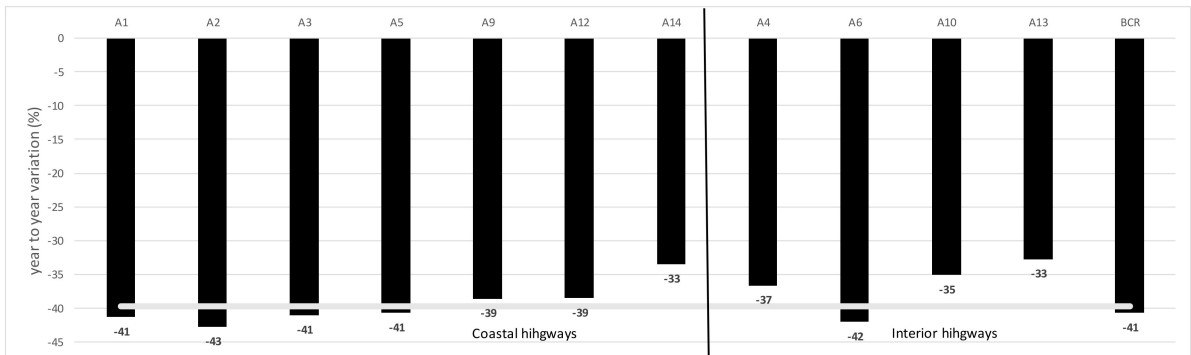

**Figure 3.** March variation of traffic (year-on-year variation). Source: authors, based on data from Brisa. Note: average reduction is 40%.

The variation of traffic levels for April, which was the peak month of the crisis, is presented in Figure 4. This also confirms our third hypothesis. The reduction in traffic is swift at the begging of the lockdown, but recovery is substantially slow. The total reduction in the level of traffic was around 70%. The pattern of changes in traffic levels between coastal and interior highways is similar to that of March, registering a 72% reduction in coastal highways and a 68% (66% if the A6 is excluded) reduction in interior highways. Turning now to May (Figure 5), the average reduction in highway traffic levels was 48%, representing an improvement compared with April of the same year (which registered a significant reduction of more than 80%). Coastal highways registered a year-on-year reduction of 50% in May, with the equivalent reduction in the interior highways (41% if the A6 is excluded). Finally, the data for June is presented in Figure 6, where it can be seen that the total reduction in levels of traffic was around 27% (which represents an increase of 43% when compared with the previous month of May). This total for June can be broken down to a reduction in traffic levels in coastal highways of 29% and of 25%for the interior (22% if the A6 is excluded). This presents evidence of our fourth hypothesis that the reduction in traffic is faster in urban areas, and the recovery is slower in such areas. Eisenhardt (1991) referred to when theory and data match closely, and the theory is empirically valid.

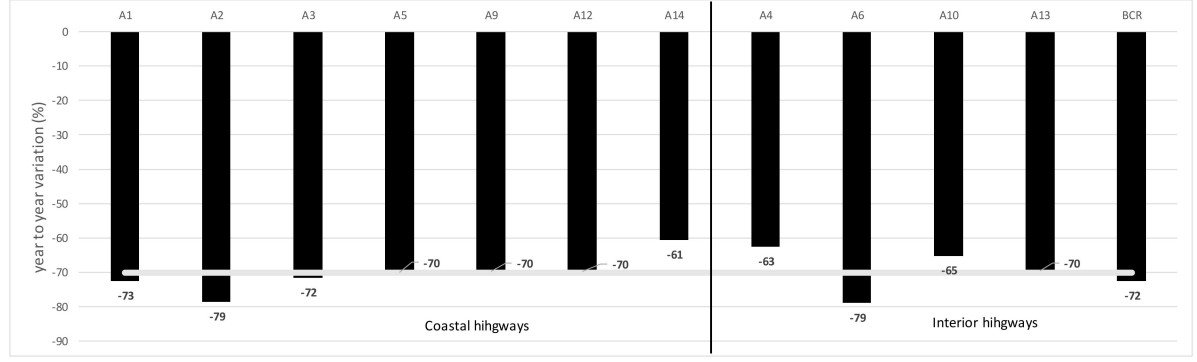

**Figure 4.** April variation of traffic (year-on-year variation). Source: authors, based on data from Brisa. Note: average reduction is 70%.

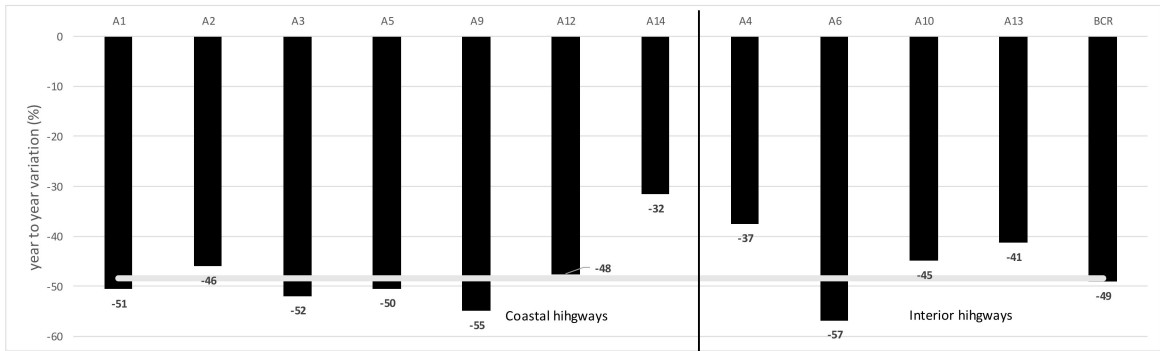

**Figure 5.** May variation of traffic (year-on-year variation). Source: authors, based on data from Brisa. Note: average reduction is 48%.

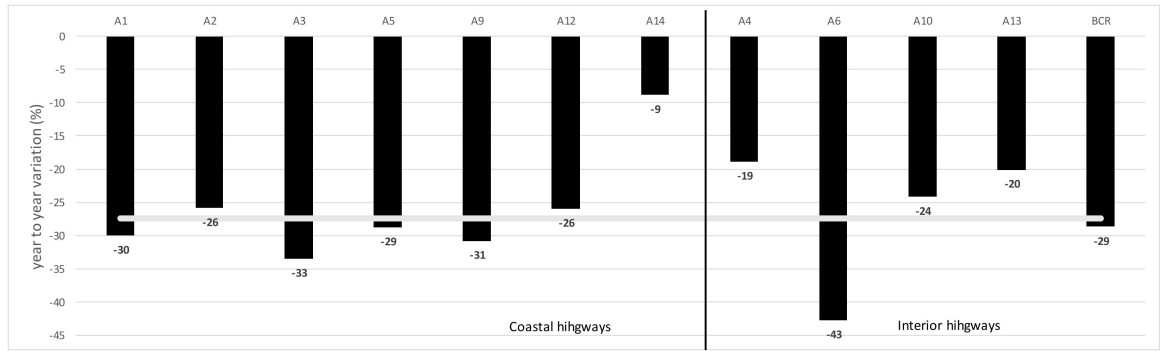

**Figure 6.** June variation of traffic (year-on-year variation). Source: authors, based on data from Brisa. Note: average reduction is 27%.

## 5. Discussion Section and Interdisciplinary Contribution

The disruptive pandemic event tested highway concession operators' ability to cope with uncertainty and unforeseen interdisciplinary challenges. The impact of COVID-19 on road concessions management and operation is a paradigmatic case of interdisciplinarity. There are four main areas in the intersection—health and safety, engineering, business and management and economic impact—all of them interrelated.

Table 3 provides an overview of the case study's main findings in each of these four areas while extracting policy implications that can help highway concessions operators in future disruptive events. [29] argued that similar pandemic events could likely happen in the future, and the lessons learnt with COVID-19 will undoubtedly be of great importance.

**Table 3.** Interdisciplinary nature of the study.

|  | Health and Safety | Engineering | Business and Management | Economic Impact |
|---|---|---|---|---|
| Case study findings | Clear identification of activities able to telework (essential financial and HR) Establish clear workflows and implement the H&S measures to decrease risk | No significant impact on maintenance activities Maintenance was able to continue *quasi* normal operation without impact on infrastructure quality | Significant impact on revenues Still not clear the full extent of impact The extension of impact will depend on the speed of economic (and traffic recovery) | Higher impact on light-weight vehicles Higher impact on urban areas and coastal areas Impact in immediate upon lockdown but recovered slowly |
| Future recommendations | Fast deployment of individual protection equipment for workers | Ensure that critical inspection and maintenance activities remain operational | Identify short term liquidity solutions Long term economic impact is still unknown | Plan the reduction in road activities such as toll collection and road assistance should based on the location of the motorway |

Overall, the main lesson is the need to quickly and decisively identify and prioritise acting in critical areas. In the case of road operators, those critical areas ensure that toll operation and road assistance remain fully operational. From a health and safety perspective, this allows them to initiate and deploy the necessary personal protection equipment is ensuring the fulfilment of health and safety standards. Secondly, although it may not seem urgent at first, it is critical to ensure that if the disruptive event will last more than a few days (for example the case of COVID-19), it is necessary to ensure that routine maintenance and inspection activities are carried out, given the proper worker's health and safety considerations. In our case study, no relevant disruption in such activities was observed, highlighting that the construction and engineering sector can continuously operate in a pandemic event. This corroborates other authors that have claimed that the construction and engineering sector was one of the least impacted by the pandemic [30].

The same does not happen from a revenue perspective. The impacts on traffic were substantial and, as such, the revenue dropped significantly. The full economic impact is still unknown. Many concessions have explicit clauses that allow for the request of economic and financial rebalance of the concession in the pandemic context (see more on transport renegotiations in [31,32]. Nevertheless, during the state of emergency, established by the government, all potential requests for renegotiations were put on hold. Such request will likely emerge in a post-pandemic context.

In terms of a more significant economic impact, it is crucial to notice that it has been asymmetrical, given the fact that, for instance, the urban areas registered more significant decreases in travel and, also, coastal areas. This is important to help future forecast disturbances, given that highway operators have different highway profiles in terms of their location. The expected reduction inactivity is not even considering the rural vs. urban, costal vs. interior or light vs. heavy vehicle, which will impact the planning and assignment of works.

## 6. Conclusions

In September 2020, the COVID-19 pandemic is still having a disruptive impact on society in Portugal. This short-term impact will likely affect daily life for many months to come—or indeed years—however, whether there will be a structural and definitive impact is still unknown.

This paper focusses on one of the most relevant infrastructures of societies—highways. Adopting an operator perspective focuses on two main dimensions—management response and traffic levels. The first dimension concerns the daily activities that comprise the operation of a highway (accident assistance, security, toll collection, or other areas of activity) and the monitoring and maintenance of the infrastructure (bridges, tunnels, slopes, pavement, among others). On the other hand, the second dimension addresses the most critical risk factor of the business of operating a highway—levels of traffic. Traffic represents revenues, bearing in mind that the highways under analysis are tolled.

Our case study shows that concerning BRISA, the management response essentially represented the first response to short term impacts. Highway operators throughout the country were hardly prepared for an event such as a worldwide pandemic, and the majority subsequently became focussed on ensuring the safety of their employees and clients, a priority to avoid service disruption.

Moving to the second dimension—levels of traffic, these have been severely affected by the pandemic situation, albeit at distinct levels regarding the different types of vehicles (and toll levels). The most significant impact was caused by the declaration of a state of emergency by the Portuguese government and the subsequent lockdown during the end of March and April. However, encouragingly, there have been some signs of recovery since then.

Despite these short-term responses and their respective impacts, a fundamental change has occurred which academics and practitioners are still trying to grasp and which undoubtedly represents an area for future research—how is the pattern of traffic changing?

In a post-COVID-19 period, will the individual use of private cars as a mode of transport regain importance, owing to the protection it offers compared with public transport? From a highway management perspective, will highway users accelerate the adoption of electronic forms of toll payment, and will manual tolling booths disappear faster than anticipated? These, and other issues, will undoubtedly stimulate a rich research agenda in the coming years.

Future studies should also provide a more vertical analysis on the subjects of worker safety and labour management. Our study was also limited by data privacy issues from the perspective of the workers and commercially sensitive data from the perspective of the firm.

**Author Contributions:** Conceptualization, J.M.S.; methodology, J.M.S. validation, J.M.S., and C.O.C.; writing—original draft preparation, J.M.S.; writing—review and editing, C.O.C. Both authors have read and agreed to the published version of the manuscript.

**Funding:** Joaquim Miranda Sarmento gratefully acknowledges the financial support received from FCT-Fundação para a Ciência e Tecnologia (Portugal), and also the national funding obtained through a research Grant (UID/SOC/04521/2020). Carlos Oliveira Cruz gratefully acknowledges the financial support received from FCT-Fundação para a Ciência e Tecnologia (Portugal), and also the national funding obtained through a research Grant (UID/04625/2020).

**Institutional Review Board Statement:** Not applicable.

**Informed Consent Statement:** Not applicable.

**Data Availability Statement:** Not applicable.

**Conflicts of Interest:** The authors declare no conflict of interest.

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
