# Peer review of "The Impact of COVID-19 on Highway Traffic and Management: The Case Study of an Operator Perspective"

_sustainability, doi:10.3390/su13095320_

Round 1

Reviewer 1 Report

This paper documented the impacts of Covid-19 pandemic in Portugal on the management of the highway operations and also on the traffic levels of different vehicle/highway classes.  The major findings reported in this paper in both the traffic-volume variation patterns and the management responses during the lock-down period in Portugal appear to follow the general trends observed in other countries.

While the quantification of specific traffic-reduction levels for various vehicle classes in Portugal during the lock-down period could be interesting findings, the authors are recommended to clarify the following points:

  • If the lock-down level was consistent throughout the 3-months period used in this study, i.e., one common-level of lock-down was applied to the entire region,
  • If the lock-down levels were varied during the study period, how different levels of lock-downs affected the traffic reduction patterns.

Reviewer 2 Report

Dear Authors,

Thank you for getting to know your work. Manuscrypt deals with the impact of the pandemic on the road transport system in Portugal. The authors presented and proved the research hypotheses, the presented results are interesting and bring new knowledge on the impact of lockdown on the organization and value of road traffic. It is a very interesting study, well prepared but requires minor editing corrections and a broader spectrum of literature.
I present my comments below:
First of all, I propose to supplement the literature, because it is too poor, and the authors have included their own citations. First of all, I suggest presenting other studies using the case study method in the transport sector. For example, I propose the following works, but please do not limit yourself to these only:

https://doi.org/10.1016/j.trpro.2019.07.209

https://doi.org/10.1515/eng-2020-0089

DOI: 10.1109/AUTOMOTIVESAFETY47494.2020.9293489

https://doi.org/10.3390/su12062295

https://doi.org/10.2478/logi-2020-0021

https://doi.org/10.2478/logi-2020-0020

https://doi.org/10.20858/sjsutst.2020.109.12

https://doi.org/10.3390/en13153869

https://doi.org/10.3390/s20061637

I suggest you put this after the 2nd sentence in the paragraph, line 72:
"The case study method is used in many scientific works concerning the transport sector, for example [...]."

In Chapter 2, I propose to insert the Portugal road map with marked main roads described in Table 1. This will be easier for readers not related to Portugal.
There are editing errors in the following lines: 247, 299, 303, 318, 323, 328, 331, 351.
Figures 1 to 5 lack a description of the axes in the diagrams, please complete this.
Please correct the literature, references are used in the journal in square brackets, otherwise it is in this manuscript.
Please correct the entry of the journal titles in the literature list, lines: 439, 440, 442, 467, 471.
Separate the literature item in verses 453-456.
I suggest completing the literature, there are 6 self-citations out of the 28th position of the reference - this is a large ratio.
Generally the work is interesting and worth improving and publishing.

Thank you.

Reviewer 3 Report

Overall, it is a good report-based case study, however, I doubt the value of the academic contribution. The hypothesis is easy to answer, and probably you can get the same findings based on whatever data you can find. Thus, I question the value of this case study to the overall research community. 

Besides that, the authors were mentioning the impacts of COVID-19 on worker safety & health, engineering & management, internal business management, and overall economic impact, I was expecting the authors could dive deep into those issues, however, it seems like the authors were just discussing them in general without much insight coming from any analysis. I would suggest the authors collect more information and connect them together. Or at least do some analysis on those that you have proposed to touch upon. 

There are also many places that lack the reference; line148, "2)" was used, then where is 1)?

Figure 1 and other figures also, "Source: Authors."? What does this mean? Class of vehicles should be placed after Table 2; 

Round 2

Reviewer 3 Report

Thank you for responding to my comments. I think you have taken my comments seriously, however, I am still not comfortable with some reference styles that you provide. For example, Source: firm website. What firm? where is the website. These are critical information, we need to make sure when other readers have a chance to take a look at your paper, they also have a chance to obtain the data source, and replicate the study results based on the same data - that is what all data science means. Thus, I would suggest the authors provide URLs to the sources or maybe supplemental documents if this article allows that. 

Author Response

Thank you for the suggestions. We have added the sources as requested by the reviewer and also provided the data as supplemental information (excel file).